# Digital Teacher Competence Frameworks Evolution and Their Use in Ibero-America up to the Year the COVID-19 Pandemic Began: A Systematic Review

**DOI:** 10.3390/ijerph192416828

**Published:** 2022-12-15

**Authors:** Camilo A. Velandia Rodriguez, Andres F. Mena-Guacas, Sergio Tobón, Eloy López-Meneses

**Affiliations:** 1Faculty of Education, Corporación Universitaria Minuto de Dios, Bogotá 111021, Colombia; 2Faculty of Education, Universidad Cooperativa de Colombia, Campus Bogotá, Bogotá 111311, Colombia; 3CIFE University Centre, Cuernavaca 62140, Mexico; 4Department of Education and Social Psychology, Pablo de Olavide University, 41013 Seville, Spain; 5Universidad ECOTEC, Km 13.5 Vía Samborondón, Guayaquil 092302, Ecuador

**Keywords:** digital competence, digital literacy, systematic review, COVID-19

## Abstract

The COVID-19 pandemic changed the way education was conducted, not only at the time when the face-to-face model was replaced by virtuality but also in the period of return to normality because the digital skills of teachers are not the same as before. Digital competency frameworks allow for assessment and comparisons between individuals and over time, so they can be used to understand the transformation that may have occurred in teachers’ digital competencies following the pandemic. This systematic literature review analyzes the competency frameworks that have been used in Ibero-America up to the year 2022, with the purpose of defining a concept foundation as an input on which to build a tool to assess digital competencies. The review was done following the pathway proposed by the PRISMA methodology between 2018 and 2022. Results show that there is no consensus or unification of the frameworks, and that there are five purposes in the research being conducted on digital competencies with publications concentrated on two of them. Interest on digital competence frameworks increased substantially in 2020.

## 1. Introduction

In 2020, COVID-19 was declared a global pandemic, which changed the practices of teachers at all levels of education. This prompted an abrupt transition to virtual classrooms [1] and revealed the shortcomings and challenges for teachers. Digital competency frameworks are a useful tool for analyzing such gaps and challenges because they assess competency development and provide common benchmarks for comparison. Although between 2021 and 2022 teachers have been returning to their activities in face-to-face mode, the development of digital competencies that accelerated due to COVID-19, is a fact.

Digital competencies in teaching are understood as the professional competences that educators need to take advantage of via digital technologies in their practice [2,3] it is necessary for optimal performance both among teachers and students [4] Despite being the focus of so much attention, this concept is, however, not necessarily new; it has evolved over the last 20 years, shifting from considering competence [2] from a more instrumental point of view towards the concept of competence as a more holistic approach: first, literacy was used as a major concept; after this, it changed to digital competence; and nowadays it is usually referred to as digital competencies in the plural, due to its complexity. Digital competences in teaching are a concept that is becoming increasingly essential as a requirement for teachers at all levels of schooling, regardless of the area of performance, the type of school, or the role played in the educational and pedagogical context. This is because all trends point to the use of digital technologies (in relation to, among other possible activities, searching for information, computer security, dissemination purposes, and creating or curating materials) that enable information to be processed, stored, and disseminated [5].

Given the existence of different digital competence frameworks, and following the recommendations provided in several of the studies selected, this literature review pretends to describe the most important frameworks to deepen their definitions identifying commonalities and differences between them, their fields, and their forms of application. The aims are to provide a reference point for the construction and adaptation of new frameworks, organizational policies, and infrastructure development [6], to understand how these frameworks are used in particular contexts, and to understand how these frameworks are used to promote the process of reflection and the use of digital competencies in order to further depth of the notion of competence in relation professional development [7]. The present paper focuses on the Ibero-American context; it is therefore relevant to inquire how much progress has been made in this region.

In the contemporary educational context, especially after the beginning of the COVID-19 pandemic, there is a need to analyze thoroughly this phenomenon of the increasingly rapid emergence of digital technologies, which has led to the transformation of traditional educative and the emergence of new ways of relating and interacting in the pedagogical scenario. This has led to the development of an important field of research that is increasingly gaining attention [8] currently referred to as “digital competence”, which is related both to the information society [2] and to the capacity to appropriate technologies to aid teaching, including the capacity to search for information [9] and the transfer of skills, while also considering age [10] and gender [11]. 

According to [12], the development of inclusive knowledge societies is based on four pillars: freedom of expression and freedom of information; universal access to information and knowledge; quality learning for all; and respect for linguistic and cultural diversity. It also refers to quality lifelong learning, which requires access to information and knowledge and full participation in society, which can transform economies and societies. Digital competencies grow in relation to the extent to which this digitality permeates different aspects, including education. Teachers assume new functions, and new professional pedagogies and methods are adopted [13] that enable barriers to be overcome and that expand the coverage and availability of educational services [12], among other benefits. This competence is uniquely complex and is considered to be more complex than any other type of competence [14], precisely because of its comprehensiveness and transversality in different dimensions, encompassing the search for, selection of, and classification of information [15] all of which lead to much more complex processes. This competence is recognized as one of the eight key competences for lifelong learning, according to the European Union [16]. According to [2] it is required to translate and reconfigure ICT in different contexts—we can no longer speak of just one type of digital competence, but of several interconnected digital competencies. Post-COVID-19, improving Digital Literacy is an urgent and alarming role for policymakers and education administrators to mitigate the potential mental health and social capital crisis [17].

Given its importance, such competence has been the subject of various studies encompassing the technological and pedagogical dimensions [18]. There are approaches regarding its conceptualization as the set of knowledge, skills and attitudes necessary today to be functional in a digital environment, in addition to the ability to transfer this ability to students [19]. However, authors, such as [20] have argued that, although there is a need for digital teacher profiling, there is no agreement on the concept itself. In this context, to improve understanding and facilitate implementation, various organizations have developed several ways of understanding this phenomenon that allow for the creation of training plans and organizational forms that enable the development of this competence among teachers, referred to as reference frameworks for digital teacher competence. 

Digital competence of teachers is a key aspect for education in the current socio digital context [21], so it is essential for teachers to have these kinds of competences [22]. According to [23] referring to the fundamental competences declared by UNESCO, digital competence is fundamental, and [24] refer to it as the most important competence for the 21st century. It involves the critical and confident use of information society technologies [25]. Ref. [26] referred to digital competence as a prerequisite for full and active incorporation into today’s information economies, as digital knowledge and competencies have gained importance in the development of society and the expectations placed on schools have increased [2]. In this context, ref. [27] recognize the need to deepen the understanding of the development of teachers’ digital competencies. The COVID-19 pandemic revealed that there are four teacher kinds: (a) the enthusiast, (b) the skeptic, (c) the pessimist, and (d) the affirmative [1].

Some authors have highlighted that is necessary to move away from the definition of “digital competence” as being related solely to technical knowledge and skills [2,13] The term must also encompass concrete situations [2] as well as considering different types of digital competence in teaching. Thus, digital competence must be related to different contexts that go beyond the training scenario and transcend the everyday scenario [4,13,28]. Ref. [13] drew attention to the evolution of this term and its passage through various conceptions over (at least) the last 20 years. Therefore, given the diversity of interpretations of the concept of digital competence, different competence frameworks have been developed [29] that delimit this competence and describe it in terms of levels of development with different levels of complexity [30] and different aspects, including the use of resources, production, and security, among others [13] in such a way that teachers can recognize their performance in an environment greatly enriched by the diversity of resources and computer media that lead to an innovative praxis [10]. Ref. [10] identified that teachers mainly develop competences only regarding communication and collaboration, leaving aside other areas included in the frameworks that may be equally important.

This paper attempts to answer the following questions:What are the purposes of competence frameworks?What are the most relevant differences and convergences between digital competence frameworks?

The above questions are intended to contribute to establish a concept base as input to build tools to assess the digital competencies that teachers were able to develop within the framework of COVID-19, as example.

## 2. Methodology

This study followed a systematic literature review methodology, understood as an observational and retrospective research design that synthesizes the results of multiple primary research studies [31]. In response to the questions formulated, following the protocol for publishing reviews and meta-analyses, according to [32], a systematic review is “the review of a clearly formulated question, which uses systematic and explicit methods to identify, select and critically appraise relevant research and to collect and analyze data extraction from the studies that are included in the review”. In principle, such reviews follow the subjective criteria of the researcher; however, to make the process more rigorous, the Cochrane Corporation designed a protocol for conducting these reviews, initially oriented to the area of health (specifically clinical studies); however, it is now common for this type of review to be carried out in other fields of knowledge, such as the education, given that its contribution is of great value in understanding the current state of the art in a particular subject by comparing and analyzing in detail research outputs that are related to each other in a way that is structured and provides added value [33]. The rigorousness of this type of review is ensured by adhering to the PRISMA _Preferred Reporting Items for Systematic Reviews and Meta-Analyses_ statement, which is a protocol with 27 criteria in a checklist (covering aspects from the title to the conclusions) that facilitates standardization in how the analysis is conducted and how the results are presented [32].

### 2.1. Literature Search Strategy

A search was carried out in the Scopus and Web of Science (WoS) databases because both databases are the most recognized in research with high standards in its editorial process, initially leading to the identification of 2140 documents using the query shown in Table 1.

### 2.2. Inclusion and Exclusion Criteria

The aim was for the studies to include the application of at least one digital competence framework for teachers and that they should be in the context of education, thus seeking to identify the areas where digital competence frameworks for teachers are most relevant. 

Frameworks encompass several categories based on different conceptions through which the development of teachers’ digital competencies is conceived and which a sense of pedagogical, social and professional development is evidenced [30,34]; however, there is no consensus on the concept of digital competencies [2] and, consequently, there are varied frameworks [3,30,35], which in turn makes them represent different uses and applications. Thus, this has been a topic that has gained special relevance in research [36], and there has been a significant increase in published studies. This trend (at the time of writing this review) is represented in Figure 1, extracted from the Scopus database, and using the following search criteria for the period 2001–2022: TITLE-ABS-KEY (“digital comp*”) OR TITLE-ABS-KEY (“digital skill*”) OR TITLE-ABS-KEY (“Digital literacy”) a difference in growth during the pandemic time is evident.

Taking into account the information in Figure 1, the search was limited from 2018, being the year in which the beginning of the trend is evidenced and until 2022, being the last two years where more than half of the articles found are located, which meet the inclusion/exclusion criteria, with this information and the criteria already described, the search was performed in Scopus and the Web of Science, finding a total of 2141 articles that meet the inclusion/exclusion criteria described in Table 2. Since the search was carried out in two different databases, the repeated articles were eliminated and then the screening continued for the final selection using the title and abstract. The research was grouped in pairs and the results were contrasted when they did not match between themselves the files selected. Finally, we proceeded to read the full text to determine the *n* = 86 all of which is summarized in Figure 2. Finally, with reading in full text, 9 papers were deleted from the results because the studies were conducted outside of Latin America: two in Poland, one in England, two in Turkey, one in Germany and one in China, and the last two were reviews of literature; that information was not in the title or abstract.

## 3. Results

Using the 77 articles selected (see Figure 2 for details of the document selection), a quantitative description of the observable data was made, such as the distribution by country in Table 3, the number of researchers per article, and the year of publication, which are described in Table 4, Table 5 and Table 6, respectively. The location of the study is relevant for the purposes of the study, as the review aims to highlight the gap between the contexts of Spain and Latin America. The identification of the date of publication reveals the research trend based on the number of studies, while the number of authors per publication indicates to some extent how research in this area is organized. Although competence frameworks have applicability in different contexts, only isolated studies have identified countries other than Spain and Portugal.

As is evident from Table 4, 42% of the studies are written by a team of three researchers, followed by four researchers (29%), two (22%), and one (3%). Table 5 shows that the year of the COVID-19 pandemic was when the most articles were published about teachers’ digital skills.

Subsequently, the purposes of the related studies were identified, revealing a great diversity, which were classified into five ad hoc categories: design and validation of a new instrument; concept of digital competence; classroom experiences; assessment of digital competence; and validation, updating, comparison, or adaptation of competence frameworks. Further details of these criteria are provided in Table 7, including each type of study classification and the associated references.

In the same vein, the type of study approach was identified in the methodology (Table 8), with quantitative studies being the most common, which is not surprising given that the main use for competence frameworks is the development of digital competence assessment processes and the characterization of populations through statistical data. In the Table 6 are the eight-framework identified; this described—a few dimensions or areas that analyze the digital competences and the number of items that in total evaluate, each area/dimension have some items.

Subsequently, a more detailed reading, first of the summaries and then of the methodological aspects, identified 8 the frameworks used in the 77 studies (Table 6). The original source was consulted to identify the most relevant features that account for the differences in structure and organization. In the year the COVID-19 pandemic was declared, only one of the competence frameworks was updated.

## 4. Discussion 

The main aim of this review has been to describe the existing digital competence frameworks, establishing comparisons between the conceptions and dimensions they address, looking for the points of theoretical and methodological convergence, and also seeking to identify how the studies that make use of one or several competence frameworks have been carried out, thus establishing the strongest lines of research and presenting suggestions for the undertaking of new research to further refine and broaden the conceptions of digital competence, in the post-pandemic period. 

### 4.1. Dimensions Analysis

Due to the important differences in the frameworks and the diversity of aspects in which they differ, the comparative analysis is presented by contrasting them with DigCompEdu, because it is the most widely used and the most comprehensive.

It is worth mentioning that in some frameworks the concept of dimensions is used, in others of areas and in others of competencies, so this is how it will be used in the following analysis.

The dimensions used by the frameworks vary; however, some of these frameworks have been inspired by previous ones. For example, the DigCompEdu is considered the European Framework for Digital Competences, recognized by the European Union, which means that some of the dimensions are shared with the Spanish frameworks, such as DIGIGLO [19] and the Common Digital Teaching Competence Framework [38]. In these three frameworks, the competencies of digital content creation, security, and problem solving are identified.

The Colombian framework is unique as it presents a dimension related to research, although some of the aspects related to it, such as creativity and innovation with digital technologies, are presented as a descriptor of digital competence in other frameworks. Similarly, the competence related to management exists as an independent dimension in the UNESCO ICT Competency Framework for Teachers, as well as in the Chilean and Colombian frameworks; in the other frameworks it is not present, although aspects related to other dimensions are mentioned.

The only dimension common to all the frameworks Is the pedagogical dimension, referring in each case to the teacher’s ability to make use of digital technologies in a way that supports students’ learning processes.

### 4.2. DigCompEdu, DIGIGLO, Intef (MCCDDD)

These frameworks have much in common because DIGIGLO and INTEG are based on DigCompEdu.

The difference between DIGIGLO and DigComEdu is that the former incorporates two more areas of analysis: Digital environment and Extrinsic digital engagement. The other six areas remain identical: Professional engagement, Digital resources, Teaching and learning, Assessment, Empowering learners and Facilitating learners’ digital competence. The form of evaluation of the competencies is the same as that proposed by DigComEdu.

The INTEF framework concentrates ”n on’y one of the six areas of dIgCompEdu, which is the development of students’ Digital Competence. The competencies it assesses in that area are also almost identical (see Table 9).

The INTEF framework uses the DigCompEdu levels A1, A2, B1, B2, C1 and C2, but associates five possible sub-levels to each, which makes it much more specific in the assessment.

### 4.3. DigCompEdu and COMDID

COMDID proposes an evaluation in four dimensions that [41] assessed and found to articulate with the six dimensions of DigCompEdu (see Table 10).

### 4.4. DigCompEdu and ICT Competences for Teachers’ Professional Development—Chile

The Chilean digital teacher competencies framework proposes five dimensions and in DigCompEdu, as mentioned above, six areas. The main differences are in the social, ethical, and legal dimension that is explicit in COMDID, but in DigCompEdu it is not so relevant. On the other hand, the areas related to students’ willingness to transform that are in DigCompEdu but have no equivalent in COMDID. Table 11 shows the comparison of the dimensions with the areas.

### 4.5. DigCompEdu and MEN

There are convergences and divergences between the two frameworks. The most notorious are: (1) the MEN’s framework values the competency of research, generation and dissemination of knowledge and the DigCompEdu does not, and (2) the DigCompEdu strives to value the competencies of the teacher based on the results achieved by the student, while the MEN’s framework focuses on what the teacher does.

Table 12 presents an analysis of divergences and convergences. The former are presented in one column for each of the competency frameworks and the latter in a single column for the two frameworks.

### 4.6. DigCompEdu and DiKoLAN

This framework does not coincide with those used in the selected studies; however, it is a proposal for the year 2022, so it was considered of interest to include it in the comparison because it is new and knowledge-related. In the DiKoLAN framework, seven competencies are evaluated, five dimensions are proposed and in the DigCompEdu six areas. The main differences are two: the simulation and modeling competency does not appear in DigCompEdu and the areas related to student transformation are not in DiKoLAN. Table 13 shows the comparison of the competencies with the areas.

In general, all digital competency frameworks for teachers have points in common. There are more coincidences than divergences in all cases. However, there are some elements that are only seen in one of the frameworks, so it would be necessary to reflect on whether they should be maintained or not:

Extrinsic digital engagement;

Management dimension;

Research and knowledge production;

Simulation and modeling;

Student empowerment area;

Area of development of students’ digital competence.

The last two have to do with the assessment of student transformation and only appear in DigCompEdu, which evidences a disinterest in the result and a greater emphasis on the teacher’s action.

In this analysis, the dimensions that make up the competence frameworks have been integrated. We have analyzed the architectures [18,68] the perceptions of the concept of digital competence in teaching [20,43] comparing some of the elements that make them up, such as age and gender [10] the diversity of dissimilar interpretations shown to exist between frameworks is striking. 

On one hand, it can be established that, despite the different efforts to integrate the concepts of digital competence, there is no consensus or unification of the frameworks, despite the fact that some frameworks are based on previous ones [8]. The definitions of digital competence are diverse, which shows the enormous complexity involved; although there are common elements, such as a pedagogical dimension and the production of content, there are other dimensions that some frameworks contemplate and others do not, such as, for example, the appropriation of institutional resources [19]. The challenge of establishing a common framework for assessing the development of teachers’ digital competencies is more urgent in post-pandemic times because the period of remote work could promote or degrade them.

Clearly, the dissemination and use of teacher digital competence frameworks is widespread in Spain. In contrast, although there are competence frameworks in Latin America, few studies have been carried out using them, and none of the studies in this review made use of Latin American competence frameworks (Chile and Colombia), despite having a solid foundation. This confirms the observation made by [8] about the low use of the competency frameworks developed in Latin America. 

## 5. Conclusions

This study has used a comparative analysis of these frameworks, which has made it possible to define similarities and differences in terms of dimensions, and to delve into the origin of some of them. Therefore, this article contributes to offering a global vision of the subject of teachers’ digital competencies before and after COVID-19, which allows for establishing a baseline for new developments that take advantage of the strengths and weaknesses of the competency frameworks known so far considering what it derived from it. In addition, this study was able to identify the different purposes of research on this topic, which makes it clear that most of the research is oriented to the use of frameworks, while a much smaller amount of research is oriented to the design and validation of instruments.

Although there are studies on teachers’ digital competencies, none have managed to demonstrate that there are significant differences; however, they do demonstrate preferences for the type of devices used to access them [95]. It is important therefore to carry out further studies comparing, for example, the differences that may exist by area in relation to teacher training and performance [42,73,97], as well as the level of schooling in which they are applied [50].

The studies reviewed in this paper are mainly quantitative because they are mostly oriented towards the assessment of teachers’ digital competencies and the characterization of populations; however, qualitative analysis would allow for other types of analysis that would strengthen the research on digital competence frameworks and the concept itself.

It is suggested that scholars undertake studies that demonstrate the relationship between teachers’ and students’ digital competencies; notably, only one such study was found in this review [102].

In 2020, more studies about teacher digital competency frameworks were published, evidence that the COVID-19 pandemic spurred interest in the topic (Figure 1). In the same year, papers were published on the five purposes defined in Table 9. However, most of them were oriented to describe classroom experiences and assessment of teachers’ digital competence.

It can be stated from this review ”hat,’In the Ibero-American context, the development of research related to teachers’ digital competencies is still insufficient. Given the conditions and training needs of teachers in the post-pandemic period, this is an area of research that should be deepened, in addition to seeking the adaptation or development of contextualized instruments that allow clear routes to be traced for the development of teachers’ digital competencies.

In all cases, the competence frameworks have related aims among which are the recognition of digital competence and its potential scope for teachers, the assessment of competence from a multidimensional viewpoint, the organization of training and competence-strengthening plans, the establishment of academic programs, and the design of learning experiences. However, although the purposes are similar, they conceive of different dimensions of competence, levels of depth, and categories for assessment; there are even important differences between the same descriptors for apparently similar categories. Coherence and consistency in these definitions can be observed in the European frameworks as they are derived from the those that are most important (UNESCO and DigCompEdu); however, those that are not similarly derived respond to different logics, and there is less consensus with respect to the other definitions.

## Figures and Tables

**Figure 1 ijerph-19-16828-f001:**
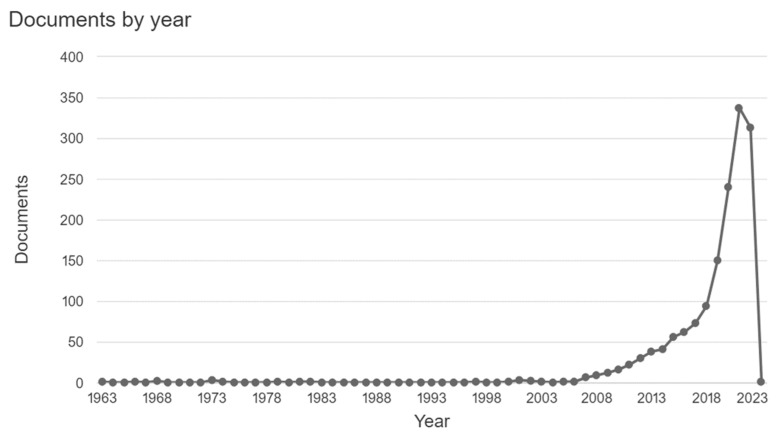
Articles found in the Scopus database (as of November 2022).

**Figure 2 ijerph-19-16828-f002:**
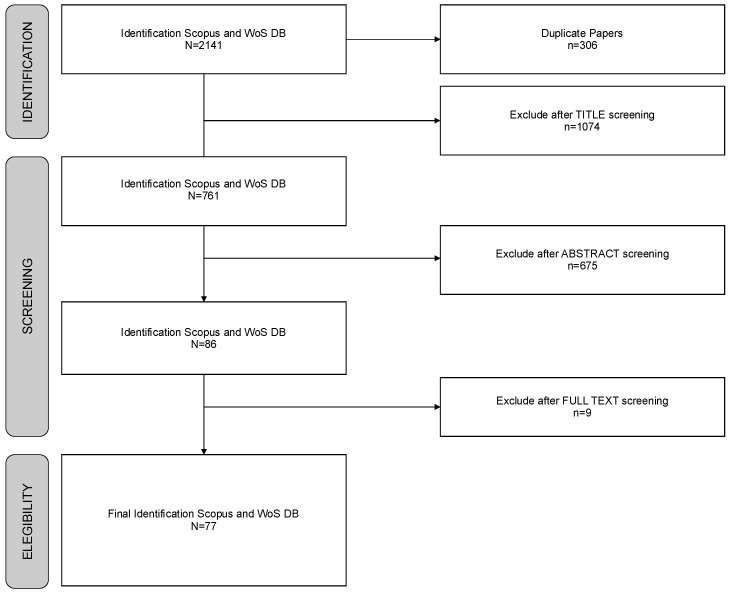
Flowchart for document selection.

**Table 1 ijerph-19-16828-t001:** Scopus and WoS search algorithms.

Scopus		WoS	
Query	# Docs	Query	# Docs
(TITLE-ABS-KEY(“digital comp*”) OR TITLE-ABS-KEY(“digital skill*”) OR TITLE-ABS-KEY(“digital literacy”) AND TITLE-ABS-KEY(“teacher*”) OR TITLE-ABS-KEY(“professor*”)) AND (LIMIT-TO (SUBJAREA,”SOCI”) OR LIMIT-TO (SUBJAREA,”ARTS”)) AND (LIMIT-TO (PUBYEAR,2022) OR LIMIT-TO (PUBYEAR,2021) OR LIMIT-TO (PUBYEAR,2020) OR LIMIT-TO (PUBYEAR,2019) OR LIMIT-TO (PUBYEAR,2018)) AND (LIMIT-TO (LANGUAGE,”English”) OR LIMIT-TO (LANGUAGE,”Spanish”) OR LIMIT-TO (LANGUAGE,”Portuguese”)) AND (LIMIT-TO (DOCTYPE,”ar”))	1091	“digital comp*” (All Fields) or “Digital Skill*” (All Fields) or “Digital literacy” (All Fields) and “teacher*” (Author) or “professor*” (Author) and 2022 or 2021 or 2020 or 2019 or 2018 (Publication Years) and Article (Document Types) and English or Spanish or Portuguese (Languages) and Social Sciences Citation Index (SSCI) (Web of Science Index) and Article (Document Types)	1050
		Grand total	2141

* We use comp* because the platform searches for any word starting with comp.

**Table 2 ijerph-19-16828-t002:** Inclusion and exclusion criteria for the review.

Inclusion	Exclusion
Empirical articles, theoretical papers Published from January 2018 to 2022 Papers published in Spanish, English and Portuguese language Studies in Social Sciences and Arts and Humanities fields Application of digital competency frameworks for teachers	Repeated Not available in full text Studies other than in the field of education

**Table 3 ijerph-19-16828-t003:** Geographical distribution of the articles included.

Country	Studies
Brazil	3
Chile	4
Colombia	2
Costa Rica	1
Ecuador	1
Spain	58
Mexico	2
Peru	6
Portugal	2
Other countries *	1
Grand total	90

* Where studies are carried out in more than one country, all countries are counted; in case the country is outside of Latin America (Australia and New Zealand), it is listed in the category “other countries”.

**Table 4 ijerph-19-16828-t004:** Number of authors per publication.

Number of Authors Per Publication	Total Number of Studies	Percentage
1	2	3%
2	17	22%
3	33	42%
4	23	29%
5	2	3%
7	1	1%
Grand total	77	100%

**Table 5 ijerph-19-16828-t005:** Distribution by year of the selected articles.

Year	Studies Included	Percentage
2018	8	10%
2019	15	19%
2020	18	23%
2021	20	26%
2022	16	17%
Grand total	77	100%

**Table 6 ijerph-19-16828-t006:** General characteristics of digital competence frameworks.

Framework	References	Country/Region	Items	No. of Dimensions	Last Updated
DigCompEdu-CheckIn	[37]	European Union	22	6	2018
DIGIGLO	[19]	Spain	29	8	2020
INTEF (MCCDDD)	[38]	Spain	21	5	2017
ICT Teaching Competence Standards	[14]	Chile		5	2011
ICT Competences for Teachers’ Professional Development	[39]	Colombia	54	5	2013
TPACK	[40]	United States	7	3	2015
COMDID-C	[41]	Spain	44	4	2019
ICT Competence Framework for Teachers	[12]	Global	18	6	2019

**Table 7 ijerph-19-16828-t007:** Ranking of the purposes of the 77 studies analyzed.

Identified Purposes	No. of Studies	References
Design and validation of a new instrument: Studies related to the validation, adaptation, translation, or updating of a digital competencies framework for teachers.	9	[35,36,41,42,43,44,45,46,47]
Concept of digital competence: Broadening the understanding of the concept of digital competence, encompassing reflections arising from the application of digital competence frameworks for teachers.	8	[20,30,48,49,50,51,52,53]
Validation, updating, comparison, or adaptation of competence frameworks. Studies related to the testing or validity of frameworks or evaluation instruments that make use of them.	13	[7,19,54,55,56,57,58,59,60,61,62,63,64]
Classroom experiences: Construction of classroom experiences and analysis of the implementation of competency frameworks. Analysis includes, for example, flipped learning	12	[65,66,67,68,69,70,71,72,73,74,75,76]
Assessment of teachers’ digital competence: Assessing teachers’ digital competence in a specific context.	35	[24,36,43,63,77,78,79,80,81,82,83,84,85,86,87,88,89,90,91,92,93,94,95,96,97,98,99,100,101,102,103,104,105,106,107]

**Table 8 ijerph-19-16828-t008:** Type of study.

Type of Study	No. of Studies	Percentage
Qualitative	7	9%
Quantitative	60	78%
Mixed	6	8%
Theoretical	4	5%
Grand Total	77	100%

**Table 9 ijerph-19-16828-t009:** Competencies assessed by the frameworks.

DigCompEdu	INTEF
Information and media literacy	Information and information literacy
Communication	Communication and collaboration
Content creation	Creation of digital content
Responsible use	Security
Troubleshooting	Troubleshooting

**Table 10 ijerph-19-16828-t010:** Articulation of COMDID with DigCompEdu.

COMDID Dimensions	DigCompEdu Areas
D1. Didactic, curricular, and methodological aspects	A3. Digital pedagogy
	A4. Evaluation and feedback
	A5. Students’ empowerment
	A6. Facilitate students’ digital competence
D2. Planning, organization and management of digital technological resources and spaces	A2. Digital resources
D3. Relational aspects, ethics, and security	A1. Professional commitment
	A5. Students’ Empowerment
	A6. Facilitate students’ digital competence
D4. Personal and professional aspects	A1. Professional commitment

**Table 11 ijerph-19-16828-t011:** Comparison of Chile and DigCompEdu framework.

Chile Framework Dimensions	DigCompEdu Areas
Pedagogical dimension	Teaching and learning area Evaluation and feedback area
Technical dimension	Digital content area
Management dimension	Does not articulate with any dimension
Social, ethical and legal dimension	Copyright issues in the digital content area
Dimension of professional development and responsibility	Area of professional commitment
There is no dimension that articulates with the areas	Student empowerment area Area of development of students’ digital competence

**Table 12 ijerph-19-16828-t012:** Divergences and convergences between DigCompEdu and the MEN framework.

DigCompEdu	MEN
The digital content area of DigCompEdu is very close to the technological competence of the MEN framework. Both are oriented to the identification, use, modification, integration, creation, and exchange of digital content for teaching; in addition, they consider the proper use of copyrights.	
The communicative competence of the MEN refers to the ability to express oneself, establish contact and relate in virtual and audiovisual spaces (MEN, 2013). The above is very close to two sub-levels of assessment of the DigCompEdu areas, specifically: Learning orientation and support (within the Teaching and Learning area) and Organizational communication (within the professional engagement area).	
The pedagogical competence of the MEN framework coincides in topics, such as the design of virtual environments and didactic strategies, autonomous learning, assessment, and collaboration, with areas 3, 4, 5 and 6 of the DigCompEdu framework.	
You have 18 possible outcomes for student assessment	There are two possible evaluation results, but not of the students, but related to the implementation of the ICT strategies and the benefit they bring to the institution’s needs.
It has a specific area for the assessment of the development of students’ digital competences, with a total of 30 possible evaluation results.	Does not value the development of digital skills by the student.
It has a specific area for the assessment of student empowerment, with a total of 18 possible evaluation results.	Does not value empowerment on the part of the student.
The research competency of the MEN framework is related to some of the competencies of the professional engagement area of DigCompEdu. Specifically with respect to the reflective attitude, participation in digital communities and the use of self-designed resources.	
It does not value the production of knowledge from research.	It includes a specific competency for research in which the development and dissemination of knowledge with 9 possible outcomes is assessed.

**Table 13 ijerph-19-16828-t013:** Comparison of DiKoLAN and DigCompEdu.

DiKoLAN Competencies	DigCompEdu Areas
Documentation	Not covered in DigCompEdu areas
Presentation	Selection and creation skills in the digital content area
Communication/collaboration	Two competencies from the area of professional engagement (organizational communication and professional collaboration) and one competency from the area of teaching and learning (collaborative learning).
Information, research, and evaluation	Digital content area selection competency Evaluation and feedback area
Data acquisition	Digital content area selection competence
Data processing	Selection competency in digital content Learning analytics competency in assessment and feedback.
Simulation and modeling	There is no area in the DigCompEdu framework.
No competencies related to the DigCompEdu areas.	Area of student empowerment Area of development of students’ digital competence

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
