# Peer review of "Digital Teacher Competence Frameworks Evolution and Their Use in Ibero-America up to the Year the COVID-19 Pandemic Began: A Systematic Review"

_ijerph, 2022, doi:10.3390/ijerph192416828_

Round 1
Reviewer 1 Report
This study reviews studies on teachers’ digital competencies and related frameworks that have been used in Ibero-America between 2018 and 2020. The structure of the manuscript accurately reflects the research aim and objectives. However, the manuscript poses several basic problems that require thorough resolution and thus a major revision to warrant its publication.
This concerns the lack of a clear delineation of the literature review from subsequent methodology and results sections and missing analyses of the dimensions of the framework (although they are mentioned in the discussion).
The comments below provide further orientation and address some additional minor points for the revision. From a language point of view, the manuscript style is suitable overall, except from some formulations (see comments below).
Abstract
P.1, line 20: “up to the year 2020”: Why limited to 2020? Because this is a very current topic, the years 2021 and 2022 should at least also yield relevant information. If for some reason you want to exclude the pandemic time from your review, please provide the corresponding rationale.
P.1, line 20: “define”: defining
P.1, lines 22-23: Please avoid brackets in the abstract, except for abbreviations.
P.1, lines 24-25: “[In] the year in which the pandemic was declared, publications concentrated on two of them.”: This information might not be useful, for example due to long review cycles in journals, which can exceed 1 year in top journals. Therefore, publications from 2020 may have been written in 2019 already.
P.1, lines 25-26: “Interest on digital 25 competence frameworks increased substantially in 2020.”: Based on what data do you make this statement? Table 6 shows a slight increase in publications, but not a very significant one (8 studies in 2018, 15 studies in 2019, 19 studies in 2020).
Introduction
P.1, lines 36-37: “the development of digital competencies that motivated COVID-19 is a fact.”: Use a different verb.
P.1, lines 37-41: This is a nice statement for the objective of your paper. However, you should place it later introduction (in the paragraph before the research questions). It usually follows from the motivation and description of the research field.
P.1, line 41: “digital resources”: Is this different from “digital competencies”? If not, please use consistent wording.
P.1, lines 42-43: “need to improve teachers’ digital competence”: plural, digital competencies
P.2, line 49: Check consistency of reference format.
P.2, lines 58-63: Please check accuracy of reference placement.
P. 2, line 63: The aims are (plural)
Literature review
P.2, line 79: 1) May be misleading, because you are doing one yourself. Maybe: theoretical background?
2) What is the objective of this section? I'm missing a clear delineation from the following sections (Methodology and Results).
P.2, line 83: “established models”: Please specify.
P.2, lines 84-89: Too many ideas in this sentence, please split it up.
P.2, line 87: “the capacity to appropriate technologies to aid teaching performance”: I am not sure what you want to say here, please check language.
P.3, lines 107-108: Which ones are these? (And why are you referring to "digital competence" in singular in most passages throughout your text?)
P.3, lines 108-110: here and throughout: check consistency of used font
P.3, line 108: “Digital Literacy”: Is this an example for a digital competence? Please define or delineate.
P.3, lines 113-116: 1) This is deviating from the definition that you gave before, indicating that there actually is no consensus regarding the definition: "Digital competence in teaching is understood as the professional competences that 44 educators need to take advantage of via digital technologies in their practice" (P.2, lines 44-45)
2) This also contradicts your general idea that there exist many "concurrent" frameworks regarding digital competencies, indicating that there is also no consensus on these
(see also last paragraph in this section).
P.3, line 135: four types: These are presumably related to the teachers?
P.3, line 144: Missing full stop
P.4, lines 163-168 and Figure 1: This belongs in the results section.
P.4, line 170: 1) Please specify why these are relevant, are these digital competence frameworks, or studies using them
2) Why is this February 2021, if the review has been submitted to this Journal at 20 October 2022?
3) This extra 1 1/2 years should make a massive difference in results, as you are showing with the trend yourself.
Methodology
P.5, line 195: This is not an algorithm. Algorithms are descriptions of procedures: "In mathematics and computer science, an algorithm [...] is a finite sequence of rigorous instructions, typically used to solve a class of specific problems or to perform a computation." (https://en.wikipedia.org/wiki/Algorithm) The sequence is missing here, these basically search terms.
P.5, Table 1: TS=((“digital comp*” OR “Digital Skill*” OR “Digital literacy”)
AND(“teacher*”OR “professor”)))): The last 2 brackets are redundant.
P.5, Lines 199-201: I'm not familiar with PRISMA. What benefit does defining PICO add to defining inclusion and exclusion criteria? Or are the latter defined according to the PICO description?
(I can see no immediate benefit, defining inclusion and exclusion criteria should be enough to make your selection process traceable.)
P.5, line 207: “The search was carried out in the WoS and Scopus databases”: Redundant, please delete.
P.6, Table 3: “Repeated”: Do you mean redundant?
Results
P.6, line 214 and P.7, Figure 2: This belongs in the Methodology section.
P.6, line 217: “purposes”: purpose, singular?
P.6, line 227: Here and throughout: Check reference errors from your text processing program (these can e.g., be seen in the finished PDF file)
P.7, Figure 2: 1) Belongs in methodology section
2) Quality of the figure is not sufficient, please use some vector graphics format, e.g., .wmf or .emf file
3) On the left-hand side, eligibility is mentioned twice. The second one should be "final sample" or something related?
4) In step 2, the "repeated" in "Review of repeated records" is confusing, because the repetition/redundancy is part of your exclusion criteria. Maybe, at this point, it is sufficient to write n=723
P.7, Table 5: Please also include “Grand total” in Table 4.
Pp. 6-9, Tables 4-9: decide on consistent wording for "Studies", "N", "items", … and use only one!
P.7, lines 238-242 and Pp.7-8, Table 7: This paragraph and Table 7 describe the underlying frameworks, instead of the "original" 42 studies of your review. Please describe this information after you describe all 42 studies, i.e., after Table 10. (inconsistent text structure)
P.7, line 239: “identified the frameworks”: Number of them is interesting à identified 8 frameworks
P.8, Table 8: Please describe in the text the meanings of the columns "Items" and "No. of dimensions" (What are items, and what are dimensions?)
P.8, Table 9, line 259: Why is this a ranking/evaluation? Maybe use the word "classification"/"categorization"?
P.8, Table 9: 1) The total of this column adds up to 41, instead of 42. So one study either has no purpose, or it is missing in the analysis.
2) Are there no studies with multiple purposes?
3) “Validation, updating, comparison, or adaptation of competence frameworks.”: This is the only category that is missing a more detailed description. Please consider writing one (for consistency)
P.10, Table 10, first column: 1) Why is this a rank? It is not determined, e.g., based on the no. of times used (or the information provided in the table is inaccurate ... digital competence is only used 3 times, higher education is used 6 times), it is also not based on alphabetical order.
2) If this ranking were based on "no. of times used", "Initial teacher training" was ranked incorrectly.
Conclusions and discussion
P.11, line 301: An open question that I had after the analysis: Can/should the dimensions and purposes of frameworks be integrated?
P.11, lines 302-303, “The main aim of this review has been to describe the existing digital competence frameworks”: What about the following purpose stated in the introduction? "with the purpose of defin[ing] a foundation on which to build a tool to assess the digital competencies that teachers were able to develop in the framework of COVID-19" (p.1, lines 39-41)
Seems to be missing here.
P.11, lines 303-304: “establishing comparisons between the conceptions and dimensions they address”: The dimensions have not been addessed in your analysis (except for their number, see Table 7, which is a very weak comparison)
P.11, lines 304-306: “and also seeking to identify how the studies that make use of one or several competence frameworks have been carried out, thus establishing the strongest lines of research”: You do not present any data/analysis on this and therefore cannot support this statement. As far as I can see from Table 9, you did analyse the purpose of studies. However, you do not present any data on which study uses which framework.
P.11, lines 306-308: “and presenting suggestions for the undertaking of new research to further refine and broaden the conceptions of digital competence, in the post-pandemic period.”: This is also missing up to this point. (Maybe, this wording is a language problem)
P.11, lines 310-311: “similarities and differences in terms of dimensions”: missing in the analysis
P.11, lines 319-325: Where is this analysis? Am I maybe missing some appendix?
P.11, line 332: Please dont write "new" results (i.e., new to the reader) in the conclusion/discussion. These belong in the "results" section.
P.12, lines 342-343: “Although there are studies on teachers’ digital competences that make use of competency frameworks to characterise populations based on their self-concept”: Confusing, what do you mean?
P.12, lines 353-359: These are results, not discussion.
P.12, lines 367-368: You have no data that supports this statement. The search terms are specifically not related to teaching at all ("TITLE-ABS-KEY (“digital comp*”) OR TITLE-ABS-167 KEY (“digital skill*”) OR TITLE-ABS-KEY (“Digital literacy”). ")
P.12, line 378: “In all cases, the competence frameworks have similar purposes”: You also do not show any data to support this statement, the purposes were related to studies, not frameworks (see Table 9).
Author Response
Dear Reviewer
Best regards,
We thank for read and suggestions to our work, in detail we read and corrected each one of them, continuing you can see the answer in the next table. We attach the document with change suggest from all reviewers and a document without change control too, to easier read it.
Abstract
|
Observation |
Answer |
|
P.1, line 20: “up to the year 2020”: Why limited to 2020? Because this is a very current topic, the years 2021 and 2022 should at least also yield relevant information. If for some reason you want to exclude the pandemic time from your review, please provide the corresponding rationale. |
Included new query until 2022 and part of 2021 |
|
P.1, line 20: “define”: defining |
Change was made |
|
P.1, lines 22-23: Please avoid brackets in the abstract, except for abbreviations. |
Brackets was deleted, they aren’t necessaries. |
|
P.1, lines 24-25: “[In] the year in which the pandemic was declared, publications concentrated on two of them.”: This information might not be useful, for example due to long review cycles in journals, which can exceed 1 year in top journals. Therefore, publications from 2020 may have been written in 2019 already. |
Dataset was actualized until 2022 by this reason the comment was withdraw |
|
P.1, lines 25-26: “Interest on digital 25 competence frameworks increased substantially in 2020.”: Based on what data do you make this statement? Table 6 shows a slight increase in publications, but not a very significant one (8 studies in 2018, 15 studies in 2019, 19 studies in 2020). |
With new query is evident the difference |
Introduction
|
Observation |
Answer |
|
P.1, lines 36-37: “the development of digital competencies that motivated COVID-19 is a fact.”: Use a different verb. |
Was changed by “development of the digital competencies that accelerate due to COVID 19 was a fact” |
|
P.1, lines 37-41: This is a nice statement for the objective of your paper. However, you should place it later introduction (in the paragraph before the research questions). It usually follows from the motivation and description of the research field. |
Paragraph was moved by the recommendation |
|
P.1, line 41: “digital resources”: Is this different from “digital competencies”? If not, please use consistent wording. |
They are different; Digital resources refer to tools that teachers had available and knew, but digital skills are more complex, involve more than tools |
|
P.1, lines 42-43: “need to improve teachers’ digital competence”: plural, digital competencies |
Was corrected |
|
P.2, line 49: Check consistency of reference format. |
Was corrected |
|
P.2, lines 58-63: Please check accuracy of reference placement. |
We improve the redaction |
|
P. 2, line 63: The aims are (plural) |
Was corrected |
|
P.2, line 79: 1) May be misleading, because you are doing one yourself. Maybe: theoretical background? |
Text added: If is compared with the number of studies identified in Spain |
Literature review
|
Observation |
Answer |
|
2) What is the objective of this section? I'm missing a clear delineation from the following sections (Methodology and Results). |
This section we fusioned with the introduction, for avoid some confusion |
|
P.2, line 83: “established models”: Please specify. |
Was changed for “traditional educative models” |
|
P.2, lines 84-89: Too many ideas in this sentence, please split it up. |
Was separated |
|
P.2, line 87: “the capacity to appropriate technologies to aid teaching performance”: I am not sure what you want to say here, please check language. |
The “performance” word was deleted. We refer to use of technologies for improve teaching |
|
P.3, lines 107-108: Which ones are these? (And why are you referring to "digital competence" in singular in most passages throughout your text?) |
Some authors use singular like a general concept; however, we corrected in this document using plural |
|
P.3, lines 108-110: here and throughout: check consistency of used font |
Was corrected |
|
P.3, line 108: “Digital Literacy”: Is this an example for a digital competence? Please define or delineate |
Text Added: “First, was used literacy as a major concept, after this changed for digital competence and nowadays usually referred like digital competencies” |
|
P.3, lines 113-116: 1) This is deviating from the definition that you gave before, indicating that there actually is no consensus regarding the definition: "Digital competence in teaching is understood as the professional competences that educators need to take advantage of via digital technologies in their practice" (P.2, lines 44-45) |
We changed the redaction, attending your recommendation. It’s a contradiction from the original papers and we need highlight it |
|
2) This also contradicts your general idea that there exist many "concurrent" frameworks regarding digital competencies, indicating that there is also no consensus on these (see also last paragraph in this section). |
We changed the redaction, attending your recomendation |
|
P.3, line 135: four types: These are presumably related to the teachers? |
Yes, we improved the redaction |
|
P.3, line 144: Missing full stop |
Was corrected |
|
P.4, lines 163-168 and Figure 1: This belongs in the results section. |
Was moved |
|
P.4, line 170: 1) Please specify why these are relevant, are these digital competence frameworks, or studies using them |
This is an-error of translation, when was changed the graph changed the title. |
|
2) Why is this February 2021, if the review has been submitted to this Journal at 20 October 2022? |
Dataset was actualized to 2022 |
|
3) This extra 1 1/2 years should make a massive difference in results, as you are showing with the trend yourself. |
Was corrected with the actualization |
Methodology
|
Observation |
Answer |
|
P.5, line 195: This is not an algorithm. Algorithms are descriptions of procedures: "In mathematics and computer science, an algorithm [...] is a finite sequence of rigorous instructions, typically used to solve a class of specific problems or to perform a computation." (https://en.wikipedia.org/wiki/Algorithm) The sequence is missing here; these basically search terms. |
Algorithm term was replaced by Query |
|
P.5, Table 1: TS=((“digital comp*” OR “Digital Skill*” OR “Digital literacy”) AND(“teacher*”OR “professor”)))): The last 2 brackets are redundant. |
Query was changed by the actualization |
|
P.5, Lines 199-201: I'm not familiar with PRISMA. What benefit does defining PICO add to defining inclusion and exclusion criteria? Or are the latter defined according to the PICO description? |
Table was deleted joint to text |
|
(I can see no immediate benefit, defining inclusion and exclusion criteria should be enough to make your selection process traceable.) |
Was deleted |
|
P.5, line 207: “The search was carried out in the WoS and Scopus databases”: Redundant, please delete. |
Was deleted |
|
P.6, Table 3: “Repeated”: Do you mean redundant? |
Previous table was deleted, now this is the number 2 |
Results
|
Observation |
Answer |
|
P.6, line 214 and P.7, Figure 2: This belongs in the Methodology section. |
Corrected |
|
P.6, line 217: “purposes”: purpose, singular? |
Corrected |
|
P.6, line 227: Here and throughout: Check reference errors from your text processing program (these can e.g., be seen in the finished PDF file) |
Corrected, were a problem with the automatic table reference |
|
P.7, Figure 2: 1) Belongs in methodology section |
Was moved |
|
2) Quality of the figure is not sufficient, please use some vector graphics format, e.g., .wmf or .emf file |
Was replaced |
|
3) On the left-hand side, eligibility is mentioned twice. The second one should be "final sample" or something related? |
The graph was changed |
|
4) In step 2, the "repeated" in "Review of repeated records" is confusing, because the repetition/redundancy is part of your exclusion criteria. Maybe, at this point, it is sufficient to write n=723 |
Reviewed in the graph |
|
P.7, Table 5: Please also include “Grand total” in Table 4. |
Realized |
|
Pp. 6-9, Tables 4-9: decide on consistent wording for "Studies", "N", "items", … and use only one! |
Was changed |
|
P.7, lines 238-242 and Pp.7-8, Table 7: This paragraph and Table 7 describe the underlying frameworks, instead of the "original" 42 studies of your review. Please describe this information after you describe all 42 studies, i.e., after Table 10. (Inconsistent text structure) |
Was moved |
|
P.7, line 239: “identified the frameworks”: Number of them is interesting à identified 8 frameworks |
Added |
|
P.8, Table 8: Please describe in the text the meanings of the columns "Items" and "No. of dimensions" (What P.8, Table 9, line 259: Why is this a ranking/evaluation? Maybe use the word "classification"/"categorization"? are items, and what are dimensions?) |
Ok Included |
|
P.8, Table 9: 1) The total of this column adds up to 41, instead of 42. So one study either has no purpose, or it is missing in the analysis. |
Corrected with the actualization |
|
2) Are there no studies with multiple purposes? |
We review and didn’t find various purposes in neither paper. |
|
3) “Validation, updating, comparison, or adaptation of competence frameworks.”: This is the only category that is missing a more detailed description. Please consider writing one (for consistency) |
It’s added |
|
P.10, Table 10, first column: 1) Why is this a rank? It is not determined, e.g., based on the no. of times used (or the information provided in the table is inaccurate ... digital competence is only used 3 times, higher education is used 6 times), it is also not based on alphabetical order. |
Are 63 keywords analized. Was a problem with the translation, here was a numeration. We deleted the columns because didn’t offer value |
|
2) If this ranking were based on "no. of times used", "Initial teacher training" was ranked incorrectly. |
|
Conclusions and discussion
|
Observation |
Answer |
|
P.11, line 301: An open question that I had after the analysis: Can/should the dimensions and purposes of frameworks be integrated? |
We separate the discussion from the conclusions. In the discussion we add a comparative analysis of each framework with DigCompEdu which is the most used. |
|
P.11, lines 302-303, “The main aim of this review has been to describe the existing digital competence frameworks”: What about the following purpose stated in the introduction? "With the purpose of defin[ing] a foundation on which to build a tool to assess the digital competencies that teachers were able to develop in the framework of COVID-19" (p.1, lines 39-41) Seems to be missing here. |
Was corrected in the intro because is a possible use of this review but is not the central purpose. |
|
P.11, lines 303-304: “establishing comparisons between the conceptions and dimensions they address”: The dimensions have not been addressed in your analysis (except for their number, see Table 7, which is a very weak comparison) |
Analysis was added |
|
P.11, lines 304-306: “and also seeking to identify how the studies that make use of one or several competence frameworks have been carried out, thus establishing the strongest lines of research”: You do not present any data/analysis on this and therefore cannot support this statement. As far as I can see from Table 9, you did analyse the purpose of studies. However, you do not present any data on which study uses which framework. |
It's Included a short description |
|
P.11, lines 306-308: “and presenting suggestions for the undertaking of new research to further refine and broaden the conceptions of digital competence, in the post-pandemic period.”: This is also missing up to this point. (Maybe, this wording is a language problem) |
New content included |
|
P.11, lines 310-311: “similarities and differences in terms of dimensions”: missing in the analysis |
New content included |
|
P.11, lines 319-325: Where is this analysis? Am I maybe missing some appendix? |
New content included |
|
P.11, line 332: Please dont write "new" results (i.e., new to the reader) in the conclusion/discussion. These belong in the "results" section. |
We didn’t find this observation |
|
P.12, lines 342-343: “Although there are studies on teachers’ digital competences that make use of competency frameworks to characterise populations based on their self-concept”: Confusing, what do you mean? |
We improve the redaction |
|
P.12, lines 353-359: These are results, not discussion. |
Was moved |
|
P.12, lines 367-368: You have no data that supports this statement. The search terms are specifically not related to teaching at all ("TITLE-ABS-KEY (“digital comp*”) OR TITLE-ABS-167 KEY (“digital skill*”) OR TITLE-ABS-KEY (“Digital literacy”). ") |
The intention was visibilize the different approach of the frameworks, however for avoid confusion we deleted this paragraph |
|
P.12, line 378: “In all cases, the competence frameworks have similar purposes”: You also do not show any data to support this statement, the purposes were related to studies, not frameworks (see Table 9). |
The idea is important, we tried to improve the redaction. Despite their similar ideals, their approaches are different, and that is what we want to emphasize |
Thanks a lot,
Authors
Reviewer 2 Report
Dear author(s),
Thank you for the opportunity to review this paper. I agree that this is an important and pertinent topic. Although the idea is a good one, unfortunately, the way in which the study is operationalized holds back its potential contribution. There are a few areas where I would encourage the authors to give further thought, as follows:
· How and why did you choose these journals and articles?
· What was the value of IF of journals?
· Papers were independently screened by what researchers to ensure thet all relevant articles were included in this review.
· What was the level of agreement between raters?
· You need a discussion section. The discussion challenges your findings and determines the degree of compatibility with previous research.
· The discussion section needs to highlight what is new in your findings and what we can learn from a study conducted in this interesting and understudied context. Whilst the introduction sets the stage for the study by justifying the relevance of the study, the discussion is the most important section as it is in the discussion that it is all brought together, and the authors illustrates how and why the study findings advance the literature. Therefore, the discussion needs to illustrate the new insights—the contributions—in a clear and compelling manner. In other words, illustrate what we know now that we did not know before or, in effect, to clearly illustrate the contribution of the study to the different bodies of literature. Furthermore, what are the future research directions based on this new framework?
· Theoretical Contributions: Addressing all the points mentioned above will lead to a more in-depth presentation of your data which has a clearer theoretical contribution. What is the theoretical contributions?
· The authors need to draw substantive conclusions from their results, and suggest, develop recommendations for further research.
· What are the theoretical and practical implications of your study and which limitations and possible future research emerge from it?
· What are the limitations of this research and how can it be solved by other researchers?
· Using the following references could be beneficial as these add more evidence to the Methodology section:
Pret, T., & Cogan, A. (2018). Artisan entrepreneurship: a systematic literature review and research agenda. International Journal of Entrepreneurial Behavior & Research.
Hosseini, E., & Sabokro, M. (2022). A systematic literature review of the organizational voice. Iranian Journal of Management Studies, 15(2), 227-252.
Best of luck with the further development of the paper.
Result this paper: Major revision
Author Response
Dear Reviewer
Best regards,
We thank for read and suggestions to our work, in detail we read and corrected each one of them, continuing you can see the answer in the next table. We attach the document with change suggest from all reviewers and a document without change control too, to easier read it. the search was extended including new papers of 2021 and 2022 years.
|
Observation |
Answer |
|
How and why did you choose these journals and articles? |
We define some inclusion and exclusion criteria, and they are explicited. We enhance a little the description in the methodology section |
|
What was the value of IF of journals? |
All papers were found in the Scopus and WoS, didn’t was a criterion IF for selected. We consider that this is a great criterion because both are the most relevant databases with good editorial proccess |
|
Papers were independently screened by what researchers to ensure that all relevant articles were included in this review. |
All researchers participated in the selection of the papers following the PRISMA protocol and the criteria stablish |
|
What was the level of agreement between raters? |
All reviewers followed the same protocol and share his observations |
|
You need a discussion section. The discussion challenges your findings and determines the degree of compatibility with previous research. |
The section was added |
|
The discussion section needs to highlight what is new in your findings and what we can learn from a study conducted in this interesting and understudied context. Whilst the introduction sets the stage for the study by justifying the relevance of the study, the discussion is the most important section as it is in the discussion that it is all brought together, and the authors illustrates how and why the study findings advance the literature. Therefore, the discussion needs to illustrate the new insights—the contributions—in a clear and compelling manner. In other words, illustrate what we know now that we did not know before or, in effect, to clearly illustrate the contribution of the study to the different bodies of literature. Furthermore, what are the future research directions based on this new framework? |
It was added this section including a depth analysis about the dimensions/areas from the frameworks and establishing comparatives between them with the most used framework (DogCompEdu) |
|
Theoretical Contributions: Addressing all the points mentioned above will lead to a more in-depth presentation of your data which has a clearer theoretical contribution. What is the theoretical contributions? |
In conclussions we added our affirmations |
|
The authors need to draw substantive conclusions from their results, and suggest, develop recommendations for further research. |
It was added |
|
What are the theoretical and practical implications of your study and which limitations and possible future research emerge from it? |
It was added |
|
What are the limitations of this research and how can it be solved by other researchers? |
It was added |
|
Using the following references could be beneficial as these add more evidence to the Methodology section: Pret, T., & Cogan, A. (2018). Artisan entrepreneurship: a systematic literature review and research agenda. International Journal of Entrepreneurial Behavior & Research. Hosseini, E., & Sabokro, M. (2022). A systematic literature review of the organizational voice. Iranian Journal of Management Studies, 15(2), 227-252. |
We read these papers and found interesting information that we have tried to include, although they were not cited in our work. |
Thanks a lot,
Authors

Reviewer 3 Report
Introduction: The authors called this part of the paper the introduction, but it looks more like a literature review. Authors are advised to improve this section and make it more in the form of an introduction instead of constant quotations that line up one after the other.
Page 4, figure 1. Why this figure is here when the text is not related to it? This part needs reorganization
Page 5, lines 193-195: There is no justification for why authors selected this date base. Please explain
Methodology: This part seems too short and poorly described. And the methodology is the most important part of the work. In this part, the authors should ensure the replication of their methodology in future research, which is still not possible. The authors should explain how many authors searched the databases, how much agreement there is between the authors, whether they did it in pairs, groups or independently....
I suggest the authors to create a discussion in accordance with their research questions and to write an appropriate and clearly separated conclusion separately from it..
Author Response
Dear Reviewer
Best regards,
We thank for read and suggestions to our work, in detail we read and corrected each one of them, continuing you can see the answer in the next table. We attach the document with change suggest from all reviewers and a document without change control too, to easier read it.
Introduction:
|
Observation |
Answer |
|
The authors called this part of the paper the introduction, but it looks more like a literature review. Authors are advised to improve this section and make it more in the form of an introduction instead of constant quotations that line up one after the other. |
We changed this section fusioned with the review, we moved sections in other parts of the document and improving the redaction. |
|
Page 4, figure 1. Why this figure is here when the text is not related to it? This part needs reorganization |
We reorganized all the paper, including this observation. |
|
Page 5, lines 193-195: There is no justification for why authors selected this date base. Please explain |
It was added. And the search was extended including new papers of 2021 and 2022 years. |
Methodology:
|
Observation |
Answer |
|
This part seems too short and poorly described. And the methodology is the most important part of the work. In this part, the authors should ensure the replication of their methodology in future research, which is still not possible. The authors should explain how many authors searched the databases, how much agreement there is between the authors, whether they did it in pairs, groups or independently |
We tried to improve this section accepting your comments and recommendations. We improve in this section commented all the process, |
|
I suggest the authors to create a discussion in accordance with their research questions and to write an appropriate and clearly separated conclusion separately from it. |
The sections are separated and we depth in the analysis of the frameworks |
Thanks a lot,
Authors
Round 2
Reviewer 2 Report
Thank you for the opportunity to review this paper.
Reviewer 3 Report
The authors did a great improvement in their manuscript. Congratulations.